# IL-33 Inhibits TNF-α-Induced Osteoclastogenesis and Bone Resorption

**DOI:** 10.3390/ijms21031130

**Published:** 2020-02-08

**Authors:** Fumitoshi Ohori, Hideki Kitaura, Saika Ogawa, Wei-Ren Shen, Jiawei Qi, Takahiro Noguchi, Aseel Marahleh, Yasuhiko Nara, Adya Pramusita, Itaru Mizoguchi

**Affiliations:** Division of Orthodontics and Dentofacial Orthopedics, Tohoku University Graduate School of Dentistry, 4-1, Seiryo-machi, Aoba-ku, Sendai, Miyagi 980-8575, Japan; fumitoshi.ohori.t3@dc.tohoku.ac.jp (F.O.); saika.ogawa.a4@tohoku.ac.jp (S.O.); shen.wei.ren.t5@dc.tohoku.ac.jp (W.-R.S.); qi.jiawei.p8@dc.tohoku.ac.jp (J.Q.); takahiro.noguchi.r4@dc.tohoku.ac.jp (T.N.); marahleh.aseel.mahmoud.t6@dc.tohoku.ac.jp (A.M.); yasuhiko.nara.q6@dc.tohoku.ac.jp (Y.N.); adya.pramusita.q6@dc.tohoku.ac.jp (A.P.); mizo@tohoku.ac.jp (I.M.)

**Keywords:** osteoclastogenesis, osteoclast, IL-33, TNF-α

## Abstract

Interleukin (IL)-33 is a member of the IL-1 family, which acts as an alarmin. Several studies suggested that IL-33 inhibited osteoclastogenesis and bone resorption. Tumor necrosis factor-α (TNF-α) is considered a direct inducer of osteoclastogenesis. However, there has been no report regarding the effect of IL-33 on TNF-α-induced osteoclastogenesis and bone resorption. The objective of this study is to investigate the role of IL-33 on TNF-α-induced osteoclastogenesis and bone resorption. In an in vitro analysis of osteoclastogenesis, osteoclast precursors, which were derived from bone marrow cells, were treated with or without IL-33 in the presence of TNF-α. Tartrate-resistant acid phosphatase (TRAP) staining solution was used to assess osteoclast formation. In an in vivo analysis of mouse calvariae, TNF-α with or without IL-33 was subcutaneously administrated into the supracalvarial region of mice daily for 5 days. Histological sections were stained for TRAP, and osteoclast numbers were determined. Using micro-CT reconstruction images, the ratio of bone destruction area on the calvariae was evaluated. The number of TRAP-positive cells induced by TNF-α was significantly decreased with IL-33 in vitro and in vivo. Bone resorption was also reduced. IL-33 inhibited IκB phosphorylation and NF-κB nuclear translocation. These results suggest that IL-33 inhibited TNF-α-induced osteoclastogenesis and bone resorption.

## 1. Introduction

Osteoclasts are large multinucleated cells responsible for bone resorption, which are formed by differentiation and fusion of precursor cells derived from a monocyte/macrophage lineage [1]. Receptor activator of nuclear factor kappa-B ligand (RANKL) and macrophage colony stimulating factor (M-CSF) are two critical cytokines for osteoclastogenesis [2]. Tumor necrosis factor-α (TNF-α) is a key cytokine in inducing bone resorption in rheumatoid arthritis [3]. TNF-α and RANKL have a synergistic effect on osteoclastogenesis [4]. TNF-α alone can also promote osteoclastogenesis by directly stimulating the precursor cells both in vitro [5,6,7] and in vivo [8,9].

The cytokine interleukin (IL)-33 is a member of the IL-1 family [10]. IL-33 is produced as a protein of approximately 30 kDa. The receptor for IL-33 is ST2, which is expressed on the surface of various cell types [11]. IL-33 is reported to be a nuclear factor, and it is released from the nucleus as an alarmin to initiate immune responses when necrosis occurs induced by infection or by physical or chemical damage [12,13]. The immune system is able to regulate bone remodeling, and there are several reports regarding IL-33 in this context. In an earlier study, IL-33 was reported to induce osteoclast formation from CD14-positive monocytes and bone resorption [14]. By contrast, IL-33 was reported to have no effect on human CD14-positive osteoclast precursors nor any direct effect on bone remodeling [15]. It is known that IL-33 is expressed in osteoblasts [15,16]. The number of osteoclasts was significantly decreased in the transgenic mouse model overexpressing IL-33 in osteoblasts [17]. Other research groups have shown that IL-33 inhibits osteoclastogenesis from osteoclast precursors in the presence of RANKL and M-CSF [18,19]. Using human TNF-α transgenic mice, which develop spontaneous joint inflammation and cartilage destruction, IL-33 inhibited TNF-α-mediated bone loss via the IL-33/ST2 axis [20]. In a study of orthodontic tooth movement, ST2-deficient mice displayed increased mechanical loading-induced osteoclast formation and bone loss, and thus greater tooth movement [21].

Although IL-33 is likely to inhibit osteoclastogenesis, the exact mechanism remains unclear. In the present study, we performed osteoclast formation using TNF-α, and investigated whether IL-33 inhibits TNF-α-induced osteoclastogenesis and bone resorption.

## 2. Results

### 2.1. IL-33 Inhibits RANKL- and TNF-α-Induced Osteoclast Formation In Vitro

To investigate whether IL-33 affects osteoclast precursors directly, we analyzed the effects of IL-33 on RANKL- and TNF-α-induced osteoclastogenesis of osteoclast precursors. A large number of tartrate-resistant acid phosphatase (TRAP)-positive cells were observed among osteoclast precursors treated with M-CSF and RANKL/TNF-α, whereas this was significantly lower among osteoclast precursors when additionally treated with IL-33 (Figure 1a,b). Furthermore, IL-33 reduced resorption pits compared with TNF-α-stimulated osteoclast precursors (Figure 1c). Real-time reverse transcription polymerase chain reaction (RT-PCR) results also revealed that the TRAP mRNA expression level was significantly higher in the TNF-α-stimulated group compared with the other groups (Figure 1d).

### 2.2. IL-33 Inhibits TNF-α-Induced Osteoclast Formation In Vivo

To determine whether IL-33 inhibits TNF-α-induced osteoclast formation in vivo, TNF-α was injected subcutaneously into the supracalvarial region of mice for 5 days with or without IL-33. When TNF-α was administered, the number of TRAP-positive cells in the suture of histological sections was significantly increased. By contrast, the number of TRAP-positive cells was decreased in the TNF-α plus IL-33 group compared with TNF-α alone group (Figure 2a). The percentage of bone marrow space interface covered by osteoclasts was significantly higher in TNF-α. The number of TRAP-positive cells per millimeter of interface of bone marrow space was also significantly higher in TNF-α (Figure 2b). Moreover, real-time RT-PCR results also revealed that the TRAP mRNA expression level was significantly higher in the TNF-α alone group compared with the other groups (Figure 2c).

### 2.3. Inhibitory Effect of IL-33 on TNF-α-Induced Bone Resorption In Vivo

Calvariae of all mouse groups were scanned by microfocus computed tomography (micro-CT), and bone resorption was analyzed. The ratio of bone resorption area to total area in the TNF-α alone group was significantly higher than the phosphate-buffered saline (PBS) alone and IL-33 alone groups. Co-application of TNF-α and IL-33 reduced the ratio of bone resorption area compared with the TNF-α alone group (Figure 3a,b).

### 2.4. Inhibitory Effect of IL-33 on TNF-α-Induced Osteoclast Formation via Phosphorylation of IκB

We evaluated the molecular mechanism by which IL-33 inhibits TNF-α-induced osteoclast formation. TNF-α or TNF-α plus IL-33 were added for specific periods (0, 5, 15, 30, 60 min). When TNF-α was added, phosphorylation of MAPK increased transiently (Appendix A). IL-33 failed to inhibit phosphorylation of MAPK (Appendix A). Activation of p-IκB peaked at 5 min and gradually decreased when cells were treated TNF-α. We found a transient decrease p-IκB activation at 5 and 15 min when IL-33 was added with TNF-α (Figure 4).

### 2.5. Inhibitory Effect of IL-33 on NF-κB Activation by TNF-α in Osteoclast Precursors

To evaluate the effect of IL-33 on NF-κB nuclear translocation, we performed immunofluorescence imaging of osteoclast precursors cultured with TNF-α or TNF-α plus IL-33 (Figure 5a). This analysis revealed that the percentage of NF-κB nuclear localization increased when stimulated with TNF-α and recovered when IL-33 was added at the same time (Figure 5b).

## 3. Discussion

In the present study, we evaluated the effect of IL-33 on TNF-α-induced osteoclastogenesis and bone resorption in vitro and in vivo. In vivo, we focused on TNF-α-induced osteoclastogenesis. The number of TNF-α-induced TRAP-positive cells was significantly decreased by IL-33. Moreover, the ratio of the bone destruction area was also decreased by IL-33. Our results suggested that IL-33 inhibited TNF-α-induced osteoclastogenesis and bone resorption in vitro and in vivo. To the best of our knowledge, this is the first report to assess the inhibitory effect of IL-33 on TNF-α-induced osteoclastogenesis and bone resorption.

TNF-α plays a central role in inflammatory osteoclastogenesis. RANKL has been considered the essential factor for osteoclast differentiation since it was reported that mice with a disrupted *RANKL* gene displayed severe osteopetrosis [22]. However, it has been reported that TNF-α stimulated osteoclast differentiation in the presence of M-CSF independent of the RANKL-RANK system [6]. In the present study, we investigated the effect of IL-33 on RANKL- and TNF-α-induced osteoclastogenesis in vitro. The results showed that IL-33 inhibited RANKL- and TNF-α-induced osteoclastogenesis from bone marrow macrophages. Whereas a number of studies have examined the disease-associated functions of IL-33, including in asthma, allergy, anaphylaxis, cardiovascular disease, and the nervous system [11], few studies have focused on the role of IL-33 in bone metabolism [23]. In the present study, we cultured osteoclast precursors in the presence of M-CSF and RANKL. When we additionally included IL-33, RANKL-induced osteoclastogenesis was significantly inhibited. This result supported previous in vitro studies in which IL-33 had an inhibitory effect on RANKL-induced osteoclastogenesis [18,19,20]. However, in contrast to the report of Mun and colleagues, we did not observe any ability of IL-33 to stimulate osteoclastogenesis [14]. We can attribute this discrepancy to the method used in terms of different cells and culture conditions. Furthermore, we examined the effect of IL-33 when osteoclast precursors were cultured with M-CSF and TNF-α in the absence of RANKL. We found that TRAP-positive cells were significantly decreased when additionally including IL-33. Moreover, IL-33 decreased the surface area of resorption pits and TRAP mRNA expression when added with TNF-α. These results suggested that IL-33 inhibited TNF-α-induced osteoclastogenesis and resorption activity in vitro. It has been reported that osteoclastogenesis induced by the combination of RANKL and TNF-α was inhibited by IL-33 [20]. Our studies are the first to demonstrate that osteoclastogenesis induced by TNF-α in the absence of RANKL was inhibited by IL-33.

We have previously demonstrated that osteoclasts can be induced in mouse calvariae treated with TNF-α in vivo [24]. In the present study, we examined osteoclastogenesis in the suture of calvariae and bone resorption by subcutaneous administration of TNF-α into the mouse cranial region for 5 days. We demonstrated that IL-33 inhibited TNF-α-induced osteoclastogenesis in vivo. Furthermore, we also found an inhibitory effect of IL-33 on TNF-α-induced bone resorption. The extent of bone resorption was determined by the ratio of bone resorption area to total bone area, analyzed by three-dimensional images from micro-CT. Our data suggested that IL-33 inhibited TNF-α-induced osteoclastogenesis and bone resorption in vivo. It has been reported that transgenic mice overexpressing human TNF-α when treated with IL-33 displayed significant protection of joint architecture, with less bone erosion and decreased osteoclast formation [20]. Our results support those reported by others who similarly found that IL-33 inhibited TNF-α-mediated bone loss [20].

NF-κB is generally located in the cytoplasm of osteoclast precursors and is naturally inhibited by IκB. Binding of TNF-α to the receptor causes phosphorylation of IκB, which results in dissociation of IκB from NF-κB. The activated NF-κB is then translocated into the nucleus and activates transcription [25]. We have previously demonstrated that IL-4 inhibited three MAPK (p38, JNK, and ERK) in osteoclast precursors treated with TNF-α [26]. In the present study, we evaluated the molecular mechanism by which IL-33 inhibits TNF-α-induced osteoclast formation. TNF-α increased phosphorylation of MAPK (p38, JNK, and ERK) and IκB in osteoclast precursors. We demonstrated that IL-33 inhibited phosphorylation of IκB while failed to inhibit phosphorylation of MAPK. In addition, immunofluorescence for NF-κB confirmed that TNF-α led to increase in nuclear translocation of NF-κB, and IL-33 recovered it. These results indicate that IL-33 directly inhibits phosphorylation of IκB or IL-33 indirectly inhibits other pathways. It has been reported that IL-33 inhibited RANKL-induced osteoclast formation via the regulating Blimp-1 and IRF-8 expression [19]. Inhibitory effect of IL-33 on TNF-α-induced Osteoclast Formation may be related to other factors.

IL-33 could provide a potential target for therapeutic intervention in a wide range of diseases [27]. IL-33 is believed to be a therapeutic option for bone loss because IL-33 protected against bone loss in a rheumatoid arthritis mouse model [20]. Furthermore, mechanical loading increased the expression of IL-33 and its receptor ST2 in alveolar bone during orthodontic tooth movement [21]. We previously reported that orthodontic tooth movement was mediated by TNF-α-induced osteoclastogenesis [28,29], such that IL-33 is presumed to play an inhibitory role on TNF-α-mediated excessive bone inflammation. Several studies have reported that IL-33 inhibited osteoclastogenesis and may be a cytokine that regulates bone inflammation. However, there are currently few reports on the role of IL-33 in bone metabolism. Further studies are needed to clarify the role of IL-33.

## 4. Materials and Methods

### 4.1. Mice and Reagents

Eight-week-old male C57BL/6J mice were purchased from CLEA Japan (Tokyo, Japan). The protocols for all animal procedures were approved by the Tohoku University of Science Animal Care and Use Committee. Four mice were randomly assigned to each experimental group and were injected reagents. Recombinant mouse TNF-α was prepared in our laboratory as previously described [8]. Recombinant mouse IL-33 was purchased from BioLegend (San Diego, CA, USA).

### 4.2. Preparation of Osteoclast Precursors for Osteoclast Formation

To isolate bone marrow cells from mice, femora and tibiae were immediately dissected after euthanasia. The epiphyses of these long bones were removed using scissors, and the bone marrow was flushed out with PBS. The cell suspension was filtered through a 40-μm nylon cell strainer (Falcon, Corning, NY, USA) and centrifuged. The obtained cells were cultured in alpha-modified minimal essential medium (α-MEM) (Wako, Osaka, Japan) containing 10% fetal bovine serum (FBS), 1% penicillin-streptomycin (PS; 100 IU/mL penicillin G and 100 μg/mL streptomycin), and M-CSF for 3 days at 37 °C and under 5% CO_2_. Non-adherent cells were removed with PBS and adherent cells were collected using a trypsin-ethylenediamineteraacetic acid (EDTA) solution (Wako, Osaka, Japan). The obtained cells, bone marrow macrophages, were used as osteoclast precursors in the present study. Osteoclast precursors (5 × 10^4^ cells) were seeded in 200 μL α-MEM containing 10% FBS and 1% PS in a 96-well plate and cultured in medium containing M-CSF alone (100 ng/mL), M-CSF (100 ng/mL) plus RANKL (100 ng/mL) or TNF-α (100 ng/mL), M-CSF (100 ng/mL) plus RANKL (100 ng/mL) or TNF-α (100 ng/mL) with IL-33 (100 ng/mL), or M-CSF (100 ng/mL) with IL-33 (100 ng/mL). After 5 days, the cultured cells were fixed in 4% paraformaldehyde for 30 min and permeabilized with 0.2% Triton X-100 for 1 h at room temperature. TRAP staining solution was used to visualize osteoclasts. TRAP solution was prepared by mixing acetate buffer (pH 5.0), 50 mM sodium tartrate, naphthol AS-MX phosphate (Sigma Aldrich, Tokyo, Japan), and Fast Red Violet LB salt (Sigma Aldrich, Tokyo, Japan). TRAP-positive cells with ≥3 nuclei were considered to be osteoclasts and were counted under a light microscope [30,31]. For resorption pits assay, osteoclast precursors were cultured on Osteo Assay Plate 96 Well (Corning life Sciences, Corning, NY, USA) with M-CSF alone (100 ng/mL), M-CSF (100 ng/mL) plus TNF-α (100 ng/mL), M-CSF (100 ng/mL) plus TNF-α (100 ng/mL) with IL-33 (100 ng/mL), or M-CSF (100 ng/mL) with IL-33 (100 ng/mL). After 5 days, pit formation was evaluated. Data were expressed as resorption pits area per total surface area.

### 4.3. Histological Analysis

Previous reports have demonstrated that the daily injection of TNF-α (1.5 μg) into the supracalvarial region for 5 days effectively induced osteoclast formation in vivo [32]. We followed the same protocol, dose, and administration period in the present study. The mice were divided into four groups of four and subjected to injection of PBS, TNF-α alone (1.5 μg/day), TNF-α plus IL-33 (1.5 μg/day), or IL-33 alone (1.5 μg/day) for 5 days. The mice were sacrificed on the sixth day and the calvariae were immediately excised. The calvariae were fixed overnight in 4% paraformaldehyde at 4 °C and demineralized with 14% EDTA at room temperature for 3 days. The calvariae were placed into tissue cassettes and processed using a tissue-processor (Leica TP1020, Wetzlar, Germany). The calvariae were cut into three coronal parts and embedded in paraffin and cut into 5 μm sections using a microtome. Each group had four mice and four sections were quantified for each mouse. To confirm osteoclast formation, the sections were stained with TRAP solution, then counterstained with hematoxylin. TRAP-positive cells with ≥3 nuclei were considered to be osteoclasts. The number of TRAP-positive cells was counted and averaged within the sagittal sutures of each coronal part in all slides according to our previous report [33]. In addition, the percentage of interface of bone marrow space covered by osteoclasts and the number of osteoclast at interface of bone marrow space were measured.

### 4.4. Preparation of RNA and Real-Time RT-PCR Analysis

For in vitro experiment, osteoclast precursors were cultured on 60mm dishes in α-MEM with M-CSF alone (100 ng/mL), M-CSF (100 ng/mL) plus TNF-α (100 ng/mL), M-CSF (100 ng/mL) plus TNF-α (100 ng/mL) with IL-33 (100 ng/mL), or M-CSF (100 ng/mL) with IL-33 (100 ng/mL) for 5 days. Total RNA was obtained from lysed cells using RNeasy mini kit (Qiagen, Valencia, CA, USA). Calvariae from in vivo experiments were frozen in liquid nitrogen and individually homogenized using a Micro Smash MS-100R (TOMY SEIKO, Tokyo, Japan) in TRIzol reagent (Invitrogen, Carlsbad, CA, USA). Total RNA was extracted using an RNeasy mini kit (Qiagen, Valencia, CA, USA) according to the manufacturer’s protocol. cDNA was synthesized from each sample using 1 μg total RNA with Superscript IV Reverse Transcriptase (Invitrogen, Carlsbad, CA, USA). To assess the gene expression levels, a Thermal Cycler Dice Real Time system (Takara, Japan) was used for real-time RT-PCR. Each reaction comprised a total volume of 25 μL, containing 2 μL cDNA and a 23 μL mixture of TB Green Premix Ex Taq II (Takara, Shiga, Japan) and 50 pmol/μL primers. The PCR cycling conditions were as follows: initial denaturation stage (95 °C for 30 s), amplification stage (50 amplification cycles with each cycle composed of a denaturation step of 95 °C for 5 s and an annealing step of 60 °C for 30 s), and final dissociation stage (a cycle composed of 95 °C for 15 s, 60 °C for 30 s, and 95 °C for 15 s). Glyceraldehyde 3-phosphate dehydrogenase (GAPDH) mRNA was used to normalize the gene expression levels. The primers are listed in Table 1.

### 4.5. Micro-CT Analysis for Bone Destruction

To assess the bone destruction, calvariae fixed in 4% paraformaldehyde at 4 °C were scanned by micro-CT (ScanXmate-R090, Comscan, Kanagawa, Japan). Three-dimensional images of the calvariae were created using TRI/3D-BON64 software (RATOC System Engineering, Tokyo, Japan). The threshold was determined by CT phantom. Bone destruction area was measured around the bregma 150 pixels in the sagittal plane and 90 pixels in the coronal plane. Shaded areas of the same color density were considered bone destruction areas. The ratio of bone resorption area to total area was measured using ImageJ (NIH, Bethesda, MD, USA) [34].

### 4.6. Immunoblotting

Osteoclast precursors were cultured in 100-mm cell culture dishes (Corning, NY, USA) in serum-free α-MEM (serum starvation) for 3 h. TNF-α (100 ng/mL) or TNF-α plus IL-33 (100 ng/mL) were then added to the dishes for specific periods (0, 5, 15, 30, 60 min). Cytokine-treated osteoclast precursors were washed twice with PBS, then were lysed using radioimmunoprecipitation assay (RIPA) buffer (Millipore, Burlington, MA, USA) containing 1% protease and phosphatase inhibitor (Thermo Fisher Scientific, Rockford, IL, USA). Total protein concentrations were determined using Pierce BCA protein assay kit (Thermo Fisher Scientific, Rockford, IL, USA). Protein samples were treated with β-mercaptoethanol (Bio-Rad, Hercules, CA, USA) and laemmli sample buffer (Bio-Rad, CA, USA) and denatured at 95 °C for 5 min as a preparation for SDS-PAGE. Equal amounts of protein were loaded into gels 4–15% Mini-PROTEAN TGX Precast Gels (Bio-Rad, Hercules, CA, USA) and transferred to a PVDF Trans-Blot Turbo Transfer System (Bio-Rad, Hercules, CA, USA). The membranes were blocked in Block-Ace (DS Pharma Biomedical, Osaka, Japan) for 1 h at room temperature. The membranes were probed with the following antibodies: phospho-p38 MAPK (Thr180/Tyr182) (D3F9) XP Rabbit mAb, phospho-SAPK/JNK (Thr183/Tyr185) (98F2) Rabbit mAb, phospho-p44/42 MAPK (ERK1/2) (Thr202/Tyr204) Antibody, phospho-IκBα (Ser32) (14D4) Rabbit mAb (Cell Signaling Technology, MA, USA; 1:1000 dilution), monoclonal Anti-β-Actin antibody produced in mouse (Sigma Aldrich, MO, USA; 1:1000 dilution) overnight at 4 °C. The membranes were washed in tris buffered saline with Triton X-100 (TBS-T) and tris buffered saline (TBS), then incubated with anti-rabbit IgG HRP-linked Antibody (Cell Signaling Technology, MA, USA; 1:3000 dilution) or anti-mouse antibody (GE Healthcare, IL, USA; 1:10,000 dilution) for 1 h at room temperature. The membranes were washed in TBS-T and TBS again, then incubated with SuperSignal West Femto Maximum Sensitivity Substrate (Thermo Fisher Scientific, Rockford, IL, USA). The signals were detected by the FUSION-FX7.EDGE Chemiluminescence Imaging System (Vilber Lourmat, Collégien, France). The stained bands were quantified with Evolution Capt (Vilber Lourmat, Collégien, France) and the results expressed relative to control and normalized to β-Actin.

### 4.7. Immunofluorescence

Osteoclast precursors were cultured in 96-well plates in α-MEM containing TNF-α (100 ng/mL) or TNF-α plus IL-33 (100 ng/mL) for 1 h. The cultured cells were fixed in 4% paraformaldehyde for 15 min, permeabilized with 0.1% Triton X-100 in PBS for 15 min at room temperature, washed and blocked with 3% BSA in PBS for 1.5 h at room temperature. The cells were proved with NF-κB p65 (C-20) sc-372 rabbit polyclonal Antibody (Santa Cruz Biotechnology, Dallas, TX, USA; 1:100 dilution) in 3% BSA in PBS overnight at 4 °C. The cells were washed with PBS and incubated with Alexa Fluor 555 goat anti-rabbit IgG (Invitrogen, Carlsbad, CA, USA; 1:400 dilution) in 3% BSA in PBS for 1 h at a room temperature. The cells were washed and counterstained with 4′,6-diamidino-2-phenylindole (DAPI).

### 4.8. Statistical Analysis

All values are presented as the mean ± standard deviation. Statistical analyses were performed using Scheffe’s test. *p* < 0.05 was considered to be statistically significant.

## 5. Conclusions

We demonstrated that IL-33 directly inhibited TNF-α-induced osteoclastogenesis in vitro and TNF-α-induced osteoclastogenesis and bone resorption in vivo. Therefore, we have obtained evidence that IL-33 plays an important role on the inhibitory effect in TNF-α-induced osteoclastogenesis and bone resorption.

## Figures and Tables

**Figure 1 ijms-21-01130-f001:**
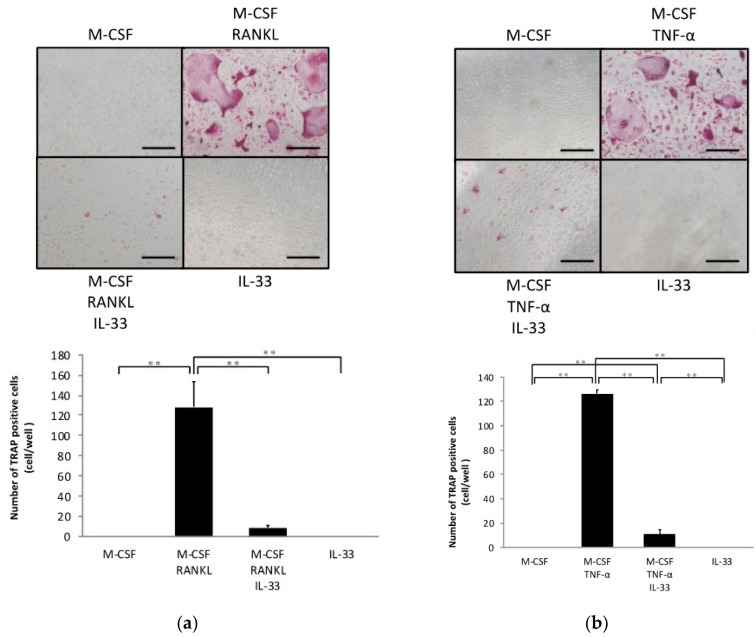
Interleukin (IL)-33 inhibited receptor activator of nuclear factor kappa-B ligand (RANKL)- and tumor necrosis factor-α (TNF-α)-induced osteoclast formation in vitro. (**a**) Microscopic images and the number of TRAP-positive cells of osteoclast precursors cultured with macrophage colony stimulating factor (M-CSF), M-CSF + RANKL, M-CSF + RANKL + IL-33, or IL-33. (**b**) Microscopic images and the number of TRAP-positive cells of osteoclast precursors cultured with M-CSF, M-CSF + TNF-α, M-CSF + TNF-α + IL-33, or IL-33. TRAP-positive cells with ≥3 nuclei were considered to be osteoclasts. (**c**) Microscopic images and the percentage of resorption pits. (**d**) Tartrate-resistant acid phosphatase (TRAP) mRNA levels of osteoclasts detected using real-time RT-PCR. Results are expressed as means ± SD. The statistical significance of differences was determined by Scheffe’s test (*n* = 4; * *p* < 0.05, ** *p* < 0.01). Scale bars = 200 µm.

**Figure 2 ijms-21-01130-f002:**
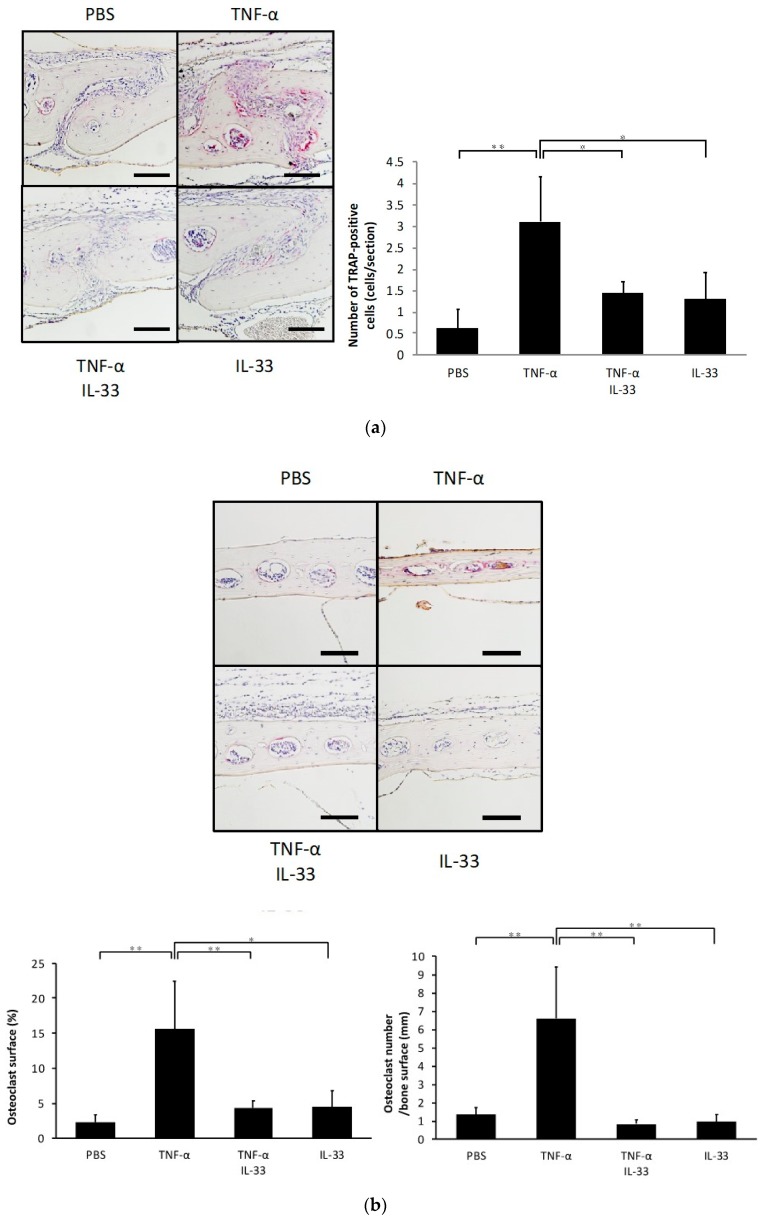
IL-33 inhibited TNF-α-induced osteoclast formation in vivo. (**a**) Microscopic images of histological sections from mouse calvariae after 5 days of daily supracalvarial administration of phosphate-buffered saline (PBS), TNF-α, TNF-α + IL-33, or IL-33. These sections were stained with TRAP solution. Hematoxylin was used as counterstaining. Scale bars = 100 µm. The number of TRAP-positive cells in the suture of calvariae among the four groups. Results are expressed as means ± SD. The statistical significance of differences was determined by Scheffe’s test (*n* = 4; * *p* < 0.05, ** *p* < 0.01). (**b**) Microscopic images of histological sections from mouse calvariae after 5 days of daily supracalvarial administration of PBS, TNF-α, TNF-α + IL-33, or IL-33. These sections were stained with TRAP solution. Hematoxylin was used as counterstaining. Scale bars = 100 µm. The percentage of interface of bone marrow space covered by osteoclast and the number of TRAP-positive cells per millimeter of interface of bone marrow space were analyzed. Results are expressed as means ± SD. The statistical significance of differences was determined by Scheffe’s test (*n* = 4; * *p* < 0.05, ** *p* < 0.01). (**c**) TRAP mRNA levels of the mouse calvariae detected using real-time reverse transcription polymerase chain reaction. Results are expressed as means ± SD. The statistical significance of differences was determined by Scheffe’s test (*n* = 3; * *p* < 0.05, ** *p* < 0.01).

**Figure 3 ijms-21-01130-f003:**
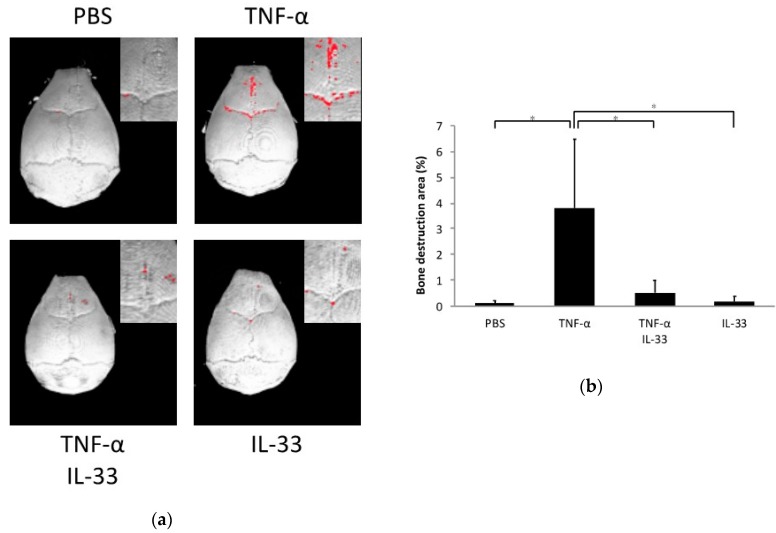
IL-33 inhibited TNF-α-induced bone resorption in vivo. (**a**) Three-dimensional images of mouse calvariae. After 5 days of daily supracalvarial injection of PBS, TNF-α, TNF-α + IL-33, or IL-33, calvariae were scanned by microfocus computed tomography (micro-CT). The red dots indicate bone destruction areas. (**b**) Ratio of the bone destruction area. Results are expressed as means ± SD. The statistical significance of differences was determined by Scheffe’s test (*n* = 4; **p* < 0.05).

**Figure 4 ijms-21-01130-f004:**
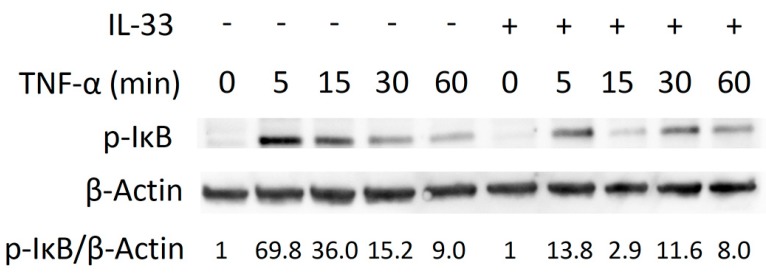
IL-33 inhibits IκB phosphorylation by TNF-α. Osteoclast precursors were exposed to TNF-α or TNF-α + IL-33 for specific periods. Cells were lysed and contents analyzed by Western blotting.

**Figure 5 ijms-21-01130-f005:**
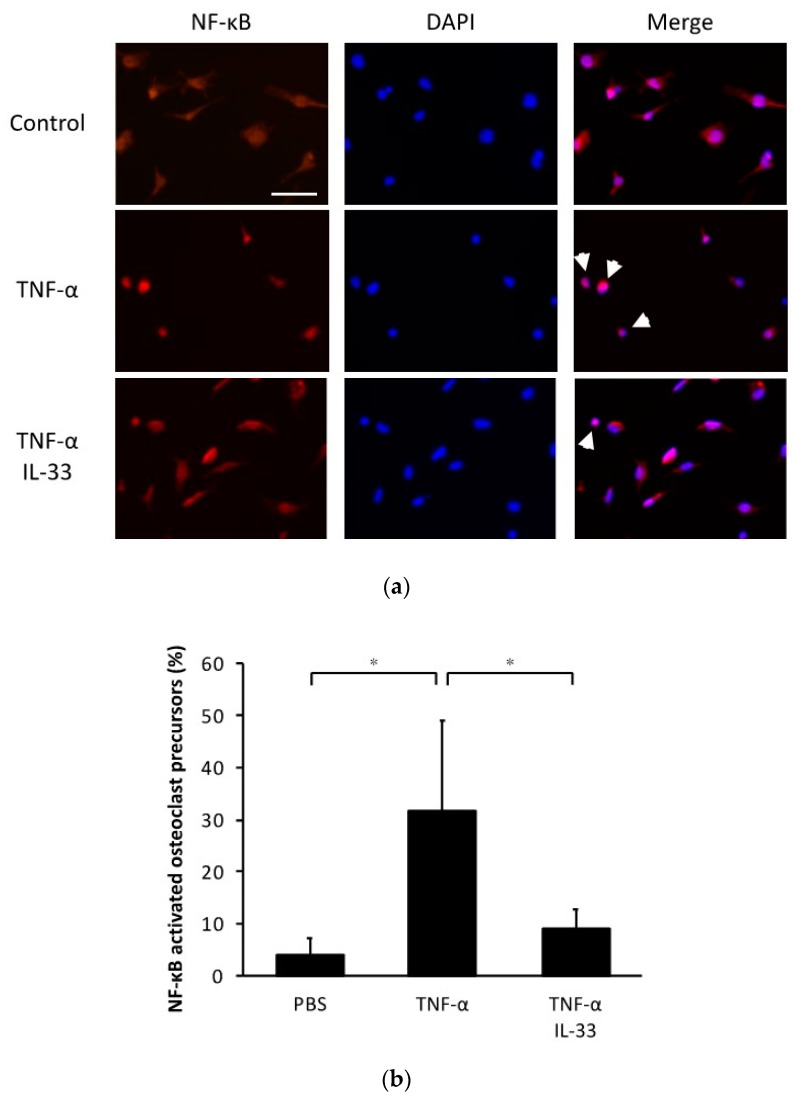
IL-33 inhibited NF-κB activation by TNF-α in osteoclast precursors. (**a**) Representative fluorescent images of osteoclast precursors cultured with TNF-α or TNF-α + IL-33 after 1 h. Cells were stained for NF-κB p65 antibody and Alexa Fluor 555. 4′,6-diamidino-2-phenylindole (DAPI) was used as counterstaining. White arrows indicate NF-κB activated osteoclast precursors. Scale bars = 50 µm. (**b**) The percentage of NF-κB activated osteoclast precursors to all osteoclast precursors. Results are expressed as means ± SD. The statistical significance of differences was determined by Scheffe’s test (*n* = 4; * *p* < 0.05).

**Table 1 ijms-21-01130-t001:** Primers used for real-time RT-PCR.

Gene	Sequence	
GAPDH	Forward	5′-GGTGGAGCCAAAAGGGTCA-3′
	Reverse	5′-GGGGGCTAAGCAGTTGGT-3′
TRAP	Forward	5′-AACTTGCGACCATTGTTA-3′
	Reverse	5′-GGGGACCTTTCGTTGATGT-3′

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
