# Peer review of "IL-33 Inhibits TNF-α-Induced Osteoclastogenesis and Bone Resorption"

_ijms, 2020, doi:10.3390/ijms21031130_

Round 1

Reviewer 1 Report

In this manuscript the authors aimed at investigating the role of Il-33 on TNFalpha-induced osteoclastogenesis and bone resorption. the authors assessed OCs formation and resorption, both "in vitro", by treating BMMs with or without IL-33 in the presence of TNFa, and "in vivo" by means of molecules injections  in the supracalvarial region.  The authors finds that IL33 inhibits TNFa induced OCgenesis and OC resorptive capacity.

This is an interesting story that, actually, could benefit of an extensive revision before beeing considerend relevant for publication, since some aspect need to be clarified.

in the results:

2.1 I think that an experiment of in vitro resorption here is needed. If the OCgenesis result will be confirmed also by pit resorption assay then the finiding could be more relevant. So please, perform OC resorption assay on bone slices or dentin discs and then stain and quantify the obtained resorption.

to enforce these data, mRNA levels of OCgenesis and Oc activity markers is needed. so please perform at leats a qPCR analysis of the different conditions of the in vitro cultures ( for example ctsk, mmp9, nfatc1, Trap etc..) 

2.2 I strongly encourage to perform again the histological analysis on the calvarial sections sice I have different concerns:

In primis, the quality of the images should be improved, the TRAP+ cells are veri hard to visulaize in PBS, TNFa+IL33 and in IL33. then, I understand that the results confirm a reduction of the OC number but I don't think it is reliable that in PBS or TNFa+Il33 or IL33 treated animals you would have 0.5 to 1.5 trap+cells/section and in TNFa treated animals 3 trap+ cells/section as means. I think these experiments should be re-done and re-analysed. Moreover I think that it is better to express these data also as OcS/BS for example or OcN/BP

2.3 for the uCT analysis, please in the methods section explain deeply how the analysis has been performed, which criteria of threshold values were considered for surface or Vox analysis.

2.5 I suggest to perform IF analysis also in Mature osteoclasts on glass and stained also with F-acting to visualize the entire cells, or even better, a differential cytoplasm/nucleus detection  for NF-B

Author Response

Thank you very much for your valuable opinion and we sincerely appreciate your valuable feedback. All the questions, we thought them over carefully. These are our responses as follows.

2.1 I think that an experiment of in vitro resorption here is needed. If the OCgenesis result will be confirmed also by pit resorption assay then the finiding could be more relevant. So please, perform OC resorption assay on bone slices or dentin discs and then stain and quantify the obtained resorption.

Response: The reviewer recommended that we should perform OC resorption assay on bone slices or dentin discs. However, we cannot get bone slices and dentin discs. Instead of them, we used Osteo Assay Plate 96 Well (Coring life Sciences, Corning, NY, USA). Osteoclast precursors were cultured on Osteo Assay Plate 96 Well with M-CSF alone (100 ng/mL), M-CSF (100 ng/mL) plus TNF-α (100 ng/mL), M-CSF (100 ng/mL) plus TNF-α (100 ng/mL) with IL-33 (100 ng/mL), or M-CSF (100 ng/mL) with IL-33 (100 ng/mL).After 5 days, pits formation was evaluated. Data were expressed as percentage of pits per field. The results showed that IL-33 reduced resorption pits compared with TNF-α-stimulated osteoclast precursors. The data show in Figure 1c.

to enforce these data, mRNA levels of OCgenesis and Oc activity markers is needed. so please perform at leats a qPCR analysis of the different conditions of the in vitro cultures ( for example ctsk, mmp9, nfatc1, Trap etc..)

Response: The reviewer recommended that we should show mRNA levels of OCgenesis and Oc activity markers. We added mRNA expression of TRAP. Osteoclast precursors were cultured on 60 mm dish withM-CSF alone (100 ng/mL), M-CSF (100 ng/mL) plus TNF-α (100 ng/mL), M-CSF (100 ng/mL) plus TNF-α (100 ng/mL) with IL-33 (100 ng/mL), or M-CSF (100 ng/mL) with IL-33 (100 ng/mL).After 5 days, we performed real-time PCR for mRNA expression of TRAP. Real-time PCR results revealed that the TRAP mRNA expression level was significantly higher in the TNF-α-stimulated group compared with the other groups. The result suggested that IL-33 inhibited TNF-a induced osteoclast formation. The result was shown in Figure 1d.

2.2 I strongly encourage to perform again the histological analysis on the calvarial sections sice I have different concerns: In primis, the quality of the images should be improved, the TRAP+ cells are veri hard to visulaize in PBS, TNFa+IL33 and in IL33. then, I understand that the results confirm a reduction of the OC number but I don't think it is reliable that in PBS or TNFa+Il33 or IL33 treated animals you would have 0.5 to 1.5 trap+cells/section and in TNFa treated animals 3 trap+cells/section as means. I think these experiments should be re-done and reanalysed. Moreover I think that it is better to express these data also as OcS/BS for example or OcN/BP.

Response: First of all, we changed images of the TRAP staining for osteoclast in Figure 2a. Next, the authors suggested that we should evaluated osteoclast formation by using other evaluation. We added evaluation of OcS/BS and OcN/BS. Microscopic images of histological sections from mouse calvariae after 5 days of daily supracalvarial administration of PBS, TNF-α, TNF-α + IL-33, or IL-33 were shown in Figure 2b. These sections were stained with TRAP solution. The percentage of interface of bone marrow space covered by osteoclast and the number of TRAP-positive cells per millimeter of interface of bone marrow space were analyzed. The percentage of bone marrow space interface covered by osteoclasts was significantly higher in TNF-α than the other groups. The number of TRAP-positive cells per millimeter of interface of bone marrow space was also significantly higher in TNF-α than the other groups. The results also suggested that IL-33 inhibited TNF-a induced osteoclast formation in vivo.

2.3 for the uCT analysis, please in the methods section explain deeply how the analysis has been performed, which criteria of threshold values were considered for surface or Vox analysis.

We added detail of uCT analysis in materials and methods section. We added “The threshold was determined by CT phantom. Bone destruction area was measured around the bregma 150 pixel in the sagittal plane and 90 pixel in the coronal plane.” in 4.5.

2.5 I suggest to perform IF analysis also in Mature osteoclasts on glass and stained also with F-acting to visualize the entire cells, or even better, a differential cytoplasm/nucleus detection for NF-B.

Response: We thank the reviewer for raising here a very important point. However, I ordered the fluorescent phalloidin, but we could not get it. It takes several weeks for it to be send from company. Therefore, we did not do this experiment in this study. In future, we will do as next study in this point.

Reviewer 2 Report

This manuscript by Fumitoshi Ohori and colleagues seeks to investigate the effect of IL-33 on osteoclast formation induced by TNFa.  Although there was some inconsistent prior literature about IL-33's effect on osteoclasts, several cited papers found that IL-33 inhibited RANKL-induced osteoclast formation.  

To address this question the authors made use of TNFa-stimulated mouse BMM cell culture system and an in vivo supracalvarial TNF-a injection experimental system.  In both of these systems they found that TNF-a induced osteoclast formation and this effect was blunted by IL-33.  In subsequent experiment investigating the molecular mechanism of this effect, they performed western blotting against commonly measured osteoclast signaling pathways- the MAPKs and NF-kB/IkB. They reported that this reduction in osteoclast formation was accompanied by a significant induction in phospho-IkB and reduced nuclear accumulation of NF-kB.  

Although the novelty of this paper is not large, it does represent a modest  step forward.  Literature cited by the authors made it highly likely that IL-33 would inhibit TNFa-induced osteoclastogenesis, but actually demonstrating this to be true was important.  The use of both in vitro cell culture at the in vivo models is a nice.  Similarly, getting at the molecular mechanisms of this effect could be significant.  Overall the writing is clear, coverage of the literature was sufficient and the discussion was fine.  If the issues raised below are addressed this will be a good study and a good fit for this journal.

Major issue:

 The western blots to establish mechanism in figure 4 are not at all convincing.  The authors report a significant difference in P-IkB induction by TNF-a in the presence of IL-33.  They seem to have come to this conclusion by separately examining P-IkB from TNF-a one one blot, TNFa+IL-33 on a different blot, and quantifying induction relative to the unstimulated negative control band on each of the blots.  These negative control lanes have close to zero signal, so small differences here can give artificially large non-real effects.  If one actually looks at the P-IkB blots side by side, there's almost no difference between them, certainly does not appear to be 10-fold difference.  This entire figure must be redone with the TNFa alone and TNFa+IL-33 lanes run side-by-side on the same gel. In figure 2, even with the higher-magnification insets the images are somewhat small and the staining intensity very weak, so it's quite difficult to see anything informative.  Additionally, they should  provide more information about how this experiment was quantified- how many slices were imaged for each mouse, and exactly how were the data from different mice combined? In figure 3, exactly how was the bone destruction area determined? Please provide more information.

Minor issues:

Although they did clearly state this in the Methods section, the figure legend, text and/or the Y-axis of the graphs in Figure 1 should also indicate somewhere that they are only counting TRAP-positive cells with 3 or more nuclei, rather than all TRAP-positive cells. Spelling errors on line 147: should be "M-CSF", line 263 "blocked", line 264 "probed"

Author Response

 Thank you very much for your helpful comments and suggestions concerning “IL-33 Inhibits TNF-α-induced Osteoclastogenesis and Bone Resorption” (manuscript ID: ijms-633695). The following are our answers to the reviewer’s comments:

Reviewer 2:

The western blots to establish mechanism in figure 4 are not at all convincing.  The authors report a significant difference in P-IkB induction by TNF-a in the presence of IL-33.  They seem to have come to this conclusion by separately examining P-IkB from TNF-a one one blot, TNFa+IL-33 on a different blot, and quantifying induction relative to the unstimulated negative control band on each of the blots.  These negative control lanes have close to zero signal, so small differences here can give artificially large non-real effects.  If one actually looks at the P-IkB blots side by side, there's almost no difference between them, certainly does not appear to be 10-fold difference.  This entire figure must be redone with the TNFa alone and TNFa+IL-33 lanes run side-by-side on the same gel. 

Response: Thank you very much for your valuable opinion and we sincerely appreciate your feedback. The reviewer recommended showing the p-IkB blots with the TNFa alone and TNFa+IL-33 lanes run side-by-side. We did this experiment on the same gel. We found a transient decrease p-IkB activation at 5 and 15 min when IL-33 was added with TNFa. These results were added in figure 4c.

 In figure 2, even with the higher-magnification insets the images are somewhat small and the staining intensity very weak, so it's quite difficult to see anything informative.  Additionally, they should  provide more information about how this experiment was quantified- how many slices were imaged for each mouse, and exactly how were the data from different mice combined?

Response: The reviewer recommended showing visible observation images in the histological section and providing more information about histological analysis. We changed the histological images to observe TRAP-positive cells in figure 2a. In addition, we added the more information in Materials and Methods section. The calvariae were cut into three coronal parts before embedded in paraffin. The mice were divided into four groups. Each group had four mice and four sections were quantified for each mouse (12 slices for each mouse). The number of TRAP-positive cells was counted and averaged. These descriptions have been added to the Materials and Methods section.

In figure 3, exactly how was the bone destruction area determined? 

Response: We added the following sentence in Materials and Methods section, “Bone destruction area was measured around the bregma 150 pixel in the sagittal plane and 90 pixel in the coronal plane.”

Although they did clearly state this in the Methods section, the figure legend, text and/or the Y-axis of the graphs in Figure 1 should also indicate somewhere that they are only counting TRAP-positive cells with 3 or more nuclei, rather than all TRAP-positive cells.

Response: We added the following sentence in figure legend, “TRAP-positive cells with ≥3 nuclei were considered to be osteoclasts.” according to the suggestion.

Spelling errors on line 147: should be "M-CSF", line 263 "blocked", line 264 "probed"

Response: We corrected these spelling errors.

Round 2

Reviewer 1 Report

The authors revised the manuscript adequately.

Author Response

The authors revised the manuscript adequately.

Response: Thank you for your comment.

Reviewer 2 Report

I had three major criticisms of the previous version of this manuscript.  Although the authors have made some revisions, these areas remain weaknesses that need to be addressed

 Size and quality of the TRAP-stained sections in figure 2 were poor. In this revision, there are now two sets of images provided in figure 2a.  It's not clear to me what's shown between the left group of images and the right group.

2.  I requested more information about how the bone destruction area was quantified in figure 3.  I'd still like a clearer explanation of how it was determined which pixels correspond to destruction versus non-destroyed area.

3.  In figure 4, I expressed significant concerns about the quantification of their Western blot data, particularly around P-IkB levels.  In response they have left in their initial analyses and added a new set of blots in 4c.  I still am highly skeptical of the quantitations in 4a-b, think that none of these data were done correctly, and would be better off removed completely.  Moreover, neither of the new blots shown in 4c actually shows any bands, and they lack quantification, so they are not at all helpful to their case either.  Ideally, the entirety of figure 4 would be redone side-by-side.

Author Response

Thank you very much for your valuable opinion and we sincerely appreciate your valuable feedback. All the questions, we thought them over carefully. These are our responses as follows.

Size and quality of the TRAP-stained sections in figure 2 were poor. In this revision, there are now two sets of images provided in figure 2a.  It's not clear to me what's shown between the left group of images and the right group.

Response: We changed the layout in Figure 2. The images in Figure 2a are for the analysis of number of TRAP-positive cells, and the images in Figure 2b are for the analysis of OcS/BS and OcN/BS.

 I requested more information about how the bone destruction area was quantified in figure 3.  I'd still like a clearer explanation of how it was determined which pixels correspond to destruction versus non-destroyed area.

Response: The reviewer recommended that we should explain about how bone destruction area was determined. We added the following sentence in Materials and Methods section, “Shaded areas of the same color density were considered bone destruction areas.”

 In figure 4, I expressed significant concerns about the quantification of their Western blot data, particularly around P-IkB levels.  In response they have left in their initial analyses and added a new set of blots in 4c.  I still am highly skeptical of the quantitations in 4a-b, think that none of these data were done correctly, and would be better off removed completely.  Moreover, neither of the new blots shown in 4c actually shows any bands, and they lack quantification, so they are not at all helpful to their case either.  Ideally, the entirety of figure 4 would be redone side-by-side.

Response: We moved Figure 2a,b to Supplementary Materials because they are negative data. In addition, we removed p-IκBdata in Figure 2a,b as advised. Next, the reviewer suggested that we should quantify new blots in Figure 4c. We added quantification in Figure 4. The result also suggested that p-IκB activation decreased at 5 and 15 min when IL-33 was added with TNF-α.